# Vitamin D and ω-3 Polyunsaturated Fatty Acids towards a Personalized Nutrition of Youth Diabetes: A Narrative Lecture

**DOI:** 10.3390/nu14224887

**Published:** 2022-11-18

**Authors:** Francesco Cadario

**Affiliations:** 1Division of Pediatrics, University of Piemonte Orientale, 28100 Novara, Italy; francesco.cadario@gmail.com; 2Diabetes Research Institute Federation, Miami, FL 33163, USA

**Keywords:** personalized nutrition, type 1 diabetes, type 2 diabetes, vitamin D, *n*-3 PUFA, *n*-6/*n*-3 PUFA ratio

## Abstract

After the discovery of insulin, nutrition has become central in the management of diabetes in order to limit glycemic rise after meals, optimize metabolic control, and prevent complications. Over the past one hundred years, international scientific societies have consecutively refined nutritional needs and optimized food intake for the treatment of diabetes. In particular, over the past century, nutrition applied with pumps for the administration of insulin and continuous glucose monitoring have allowed substantial advancement in the treatment of type 1 diabetes mellitus. The role of some substances, such as vitamin D and *n*-3 polyunsaturated fatty acids, have been proposed without univocal conclusions, individually or in combination, or in the diet, to improve the nutrition of type 1 and type 2 diabetes. This second condition, which is highly associated with overweight, should be prevented from childhood onwards. Personalized nutrition could bypass the problem, reaching a scientific conclusion on the individual subject. This article focuses on childhood and adolescent diabetes, aims to provide a narrative summary of nutrition over the past century, and promotes the concept of personalized nutrition to pediatricians and pediatric diabetologists as a possible tool for the treatment of type 1 diabetes and the prevention of type 2 diabetes.

## 1. Background

Type 1 diabetes mellitus (T1DM) is the most common endocrine disease in childhood. Its incidence is increasing in all Western countries and is showing a younger age of onset [1]. Type 2 diabetes mellitus (T2DM), until recently limited to adulthood in the Italian population, is now detectable in obese adolescents, and prediabetes (identified as high fasting blood glucose and impaired glucose tolerance) is already present in approximately 5% of obese children >10 years old [2]. T2DM incidence in adulthood is increasing in all countries and is highly associated with childhood and adolescence overweight, which in turn is on the rise. Consequently, diabetes is expected to be an unsustainable burden for health systems [3,4] and, according to the International Diabetes Federation project, by 2045, 700 million people worldwide will develop diabetes [5]. Moreover, a recent study shows that excessively high BMI in otherwise healthy adolescents is associated with T1DM appearance in early adulthood [6].

T1DM has a genetic basis related to autoimmunity, which accounts for 50–60% in the determinism, and T2DM has a polygenetic basis, which approximately contributes 50% to the disease. Because genes are expected to be stable over time, it is presumed that environmental determinants and gene–environment interactions are responsible for the increase of both diseases. Nutritional factors favoring or protecting from diabetes, overweight, obesity, and lifestyle, might act as environmental determinants. It is essential to detect each one as a protective, trigger, or accelerator factor of diabetes, because those determinants can be modifiable.

In this article, the evolution of nutrition in diabetes in the past century is summarized and light is shed on new perspectives to pediatricians and youth diabetologists regarding type 1 and 2 diabetes mellitus (excluding other forms of diabetes of childhood, which might have different claims).

## 2. Statements on Nutrition for the Care of Diabetes in Childhood and Adolescence: An Historical Overview

International Scientific Societies have provided updated guidelines for T1DM and T2DM based on evidence, which represent the state of shared knowledge on diabetes care, including lifestyle and nutrition [7,8]. Recently, the American Diabetes Association (ADA) and the European Association for the Study of Diabetes (EASD) released a consensus report on the management of T1DM in adults [9]. Most nutrition statements for T1DM and T2DM overlap in children, adolescents, and adults.

The nutritional intake of carbohydrates, fats, and proteins, which are macronutrients, are defined in Table 1 [7].

Fats should represent 30% of the total daily caloric content. The breakdown of fats should be <7–10% saturated, 10% monounsaturated fatty acids (MUFAs), and 10% polyunsaturated fatty acids (PUFAs). MUFAs are found in olive, sesame, rapeseed oils, and nuts. Source of PUFAs are corn and sunflower (*n*-6 PUFA) and fish, olive oil, and vegetables (*n*-3 PUFA). Oily marine fish, rich in *n*-3 PUFA, is recommended once or twice weekly in amounts of 80 to 120 g [7,10]. The guidelines recommend avoiding trans fatty acids as much as possible and limiting cholesterol intake to <200 mg/day. The daily protein requirement decreases in pediatric age from approximately 2 g/Kg/day during childhood to 1 g/Kg/day for a 10-year-old child and to 0.8 to 0.9 g/Kg/day in adolescence [7]. Fiber intake recommended in children is 14 g/1000 kcals (or by an alternative formula, in children >2 years old, calculated as: age in years + 5 = grams of fiber per day), up to 25–30 g/day, which represents the dose for adults [7,11].

The deficiencies of specific micronutrients, if any, must be replaced with additional intakes. Vitamin D supplement is especially recommended to athletes for optimal performance [7]. Guidelines particularly focus on diabetic dyslipidemia (identified by ADA 2021 as non-HDL cholesterol increase [8]), because it is predictive of cardiovascular disease (CVD) in adulthood [8,12]. Dyslipidemia in adolescence could be counteracted primarily through the addition of fiber in the diet [13], with regular daily physical activity [14], and then, if necessary, with drugs. One hour of aerobic physical activity per day of medium or high intensity must be prescribed to children and adolescents with diabetes [7,8,9]. The presence of obesity increases the risk of CVD, so its prevention or management is essential. Guidance for family choices regarding appropriate portion sizes, limiting the energy density of foods, meal routines, and physical activities is crucial [7,8]. Moreover, particular attention must be paid to identifying eating disorders, especially in 10–12-year-old female patients, which requires specific skills for its management. Finally, the guidelines emphasize that there is not one single type of diet for diabetes, and different kinds of diet must be considered according to family choices, ethnic habits, and religious eating standards [7,8]. Nutrition must be evaluated in the context of the overall evolution of the young for the best physical, psychic, and emotional fulfillment by overseeing weight control and blood sugar and enhancing individual choices, and it should be aimed at the whole well-being of patients.

A milestone of nutrition in T1DM has been carbohydrate counting (CHO counting), introduced shortly after the discovery of insulin, to adjust insulin doses in relation to the carbohydrate intake in each meal, thus limiting post-prandial glycemic rise. Its importance was recognized only in 1993 by the American Dietetic Association to allow variability of the diet, ensuring a good glycemic control. CHO counting must be accompanied by training in its use, because the patient has to be able to count the grams of carbohydrates contained in each meal and match insulin doses accordingly [15]. This tool has been determined for the successful use of insulin pumps. The Glycemic Index (GI, the percentage of glycemic increase determined by 100 g of a specific food vs. 100 g of sugar) and Glycemic Load (GL, the multiplication of the GI by grams of carbohydrates in a meal) are currently performed to adjust insulin requirements [16,17]. For the application of GI and GL, the interference of different foods and their preparation on the glycemic outcomes has been highlighted (a mixed meal leads to a reduction in the rise of blood glucose for a food with a high GI, and cooking increases the GI of a food, making it more quickly digestible).

One factor of the past century that has had a great impact on eating behavior is the Mediterranean Diet (MD), as theorized by Ancel and Margaret Keys in 1959 in the book “How to eat well and stay well, the Mediterranean way” [18]. It became a reference for human nutrition in the Mediterranean area and in many Western countries. The MD is based on foods whose consumption is usual in Mediterranean countries. The diet included a prevalent intake of cereals, fruit, vegetables, seeds, and olive oil; a reduction of red meat and animal fats (rich in saturated fats); and a moderate consumption of fish, white meat (poultry), legumes, eggs, and dairy products. In the Seven Countries Study in 1975, the MD took on different declinations according to the populations studied, but still had a predominance of vegetables and limited animal derivative products, and it was still combined with an active lifestyle [19]. The MD initially proposed to prevent CVD [20,21] was subsequently transposed to diabetes and obesity care [22,23], relying on the same contributions of macronutrients, fiber, fish, poultry, eggs, seeds, nuts, vegetables, and olive oil. The Mediterranean food model, not only as a diet but also as a lifestyle, has become a reference, at least in Mediterranean countries, for people with diabetes. On the 17 November 2010, the MD was recognized by UNESCO as an intangible heritage of humanity. The poor socio-economic conditions following the Second World War, which forced a limited availability of food and a sober lifestyle, undoubtedly contributed to the MD.

At the same time in England, during the Second World War, people were provided five extra welfare foods, including cod liver oil (with extra milk, eggs, meat, and orange juice). The Welfare Food scheme continued to offer supplemental cod liver oil from 1941 to 1961 as a “healthy food”, but this was discontinued due to the unfortunate outbreak of hypercalcemia caused by excessive vitamin D fortification in infant foodstuff [24]. As well as the MD, the intake of cod liver oil became in vogue in Italy, Europe, and other Western countries. In fact, it represented an additional supply of vitamins D, A, E, and *n*-3 PUFAs in the nutrition of children and adolescents. Those supplements promised many health benefits, but they were not verified, resulting in the progressive phasing out of cod liver oil daily intake as a fad. To date, cod liver oil supplementation is still used in adults, especially in higher social classes, as a more favorable eating habit within a healthy lifestyle, which makes the evaluation of its benefits for possibly bias difficult [25].

Another event related to the Second World War has been revealed, but it was not immediately recognized as relevant. Serious nutritional deprivation during pregnancy in Dutch people as a consequence of war restrictions caused restricted fetal grown and small size at birth in offspring, and increased diseases in later life. Metabolic disorders, such as glucose intolerance and obesity, an atherogenic lipid profile leading to CVD, and coagulation abnormalities, were found in this Dutch Famine Cohort [26]. For the first time, there was clear evidence of nutrient–gene interactions, supposedly through epigenetic changes.

Only in 2005 did genome-wide association studies (GWAS) provide further robust insights into the relationships between genes and environment. By comparing the single nucleotide polymorphisms (SNPs) of subjects with a specific pathology vs. an unaffected and otherwise equal population, it became possible to investigate genetic variations linked to a specific disease. Particularly in monogenic diseases, the discovery of a single mutation, for example of a protein encoding an enzyme involved in the pathogenesis of a disease, would allow the disease to be attributed to a single genetic trait. Often, SNPs associated with a disease are in non-coding regions, and these sequences probably involve genes with regulatory activities, which involve the non-Mendellian inheritance of the genetic trait. Moreover, in most cases there are multiple SNPs involved with a disease, typically in polygenic ones such as T2DM or obesity, and therefore it is difficult to understand the contribution of each SNP. In these investigations, quantitative treat locus (QLT) analysis can calculate the recurrence of SNPs associated with the specific traits of a disease, allowing determination of each contribution to the complex disease.

To date, GWAS have highlighted several relationships between nutrients and diseases, as well as the possible role of various nutrients in gene expression. This is the focus of Nutrigenomic Science, which is in constant evolution and should provide a great contribution to the knowledge of both type 1 and type 2 diabetes [27,28]. This methodological approach of investigation has provided, and will probably continue to do so in the future, further strong evidence of the role of single nutrients in the determinism and evolution of diabetes.

## 3. Micronutrients: Insight into Protective or Causative Roles in Type 1 Diabetes

In the determinism of T1DM, dietary factors acting early in life or during pregnancy supposedly play a role as environmental determinants, because autoantibodies targeting β-cells as markers of autoimmune disease may be found early in the life of affected subjects [29]. The intake of infant formula instead of breast milk, infant formula vs. highly hydrolyzed infant formula [30], early vs. late gluten introduction in the diet [31], and vitamin D administration [32] have been investigated. It is likely that other dietary factors also favor or protect infants and children from this disease, such as long-chain *n*-6 and *n*-3 PUFAs [33].

The contribution of vitamin D as a pivotal micronutrient in the diet of children and adolescents for the prevention of rickets has been extensively proven. Subsequently, vitamin D was investigated in the prevention of T1DM based on epidemiological findings, and Hyppönen et al. showed that a vitamin D supplementation (2000 IU/day) in infancy reduced the risk of T1DM by 80% later in life [32]. The hypothesized immunomodulatory action of vitamin D is largely documented [34,35], although as of today there is a lack of supporting clinical studies of a preventive role in T1DM [36].

The investigation of a causal link between T1DM and vitamin D deficiency was further addressed using data from GWAS on Mendellian traits determining the inherited unbalanced vitamin D synthesis. In a large European series, SNP-associated Mendelian mutations of vitamin D synthesis, causing low 25(OH)D level (which is the reference metabolite of vitamin D status), did not match an increased T1DM risk [37]. The Authors advised that these findings could not be generalized to non-Europeans populations, nor for 25(OH)D levels in the extremes of the normal distribution. Beyond the possible causative role of T1DM, vitamin D has also been investigated for a possibly metabolic role, because the 25(OH)D level was found to be inversely related to insulin resistance in various ethnicities [38]. Consistently, in a recent meta-analysis of 46 RCTs, vitamin D supplements conferred a metabolic advantage, suggesting that vitamin D supplementation had a positive impact on glycemic control in pediatric T1DM [39]. Moreover, Treiber et al., in a double-blind placebo RCT in 30 young patients with new-onset T1DM, found cholecalciferol supplementation (70 IU/Kg/day) improved suppressor function in regulatory T cells (T-lymphocyte subsets) [40]. The study showed a trend towards a slower decline of fasting C-peptide (marker of the endogen insulin secretion), and significantly lower insulin dose requirements after 12 months in the cholecalciferol supplemented new onsets than in a placebo group. The Authors suggested that vitamin D could act as an immunomodulatory agent in T1DM, which could possibly be useful in future combined therapies to mitigate the course of the disease.

Likewise, in healthy siblings of T1DM children, the T lymphocyte subset Th 17, a probable actor of β-cells autoimmune attack underlying the disease, was found to be increased in severe vitamin D deficiency (25(OH)D ≤ 10 ng/mL) and was reduced with cholecalciferol 1000 IU/day supplementation for 6 months [41]. These data support the idea that serum 25(OH)D deficient levels affect autoimmunity and highlight the importance of replacing vitamin D deficiency in the siblings of T1DM children.

Moreover, a case-control study of 25(OH)D levels in neonatal spots (2002–2012) from a biobank of Piedmont (North Italy) was performed between newborns that became T1DM within 10 years of age vs. controls without the disease, matched for ethnicity, gender, place, and date of birth according to the Register of T1DM of Piedmont [42]. An OR regression of disease of 0.78 (95%, CI 0.56–1.10) was found for each unit increment of log 25(OH)D. In immigrant newborn babies, the very low levels of 25(OH)D <2.14 ng/mL, indicated an increased risk of T1DM (OR 14.02, 95% CI 1.76–111.70) vs. babies with 25(OH)D ≥2.14 ng/mL, whereas no association was evident in Italians. Likewise, in two large-scale national studies in Denmark, 25(OH)D levels at birth did not involve an increased T1DM risk in the Danish population [43].

These results lead us to consider severe vitamin D deficiency as an environmental determinant of T1DM in non-Caucasian people in Italy, mostly immigrants from North Africa. Intuitively, this determinant affects subjects carrying genetical predisposition since early infancy or in pregnancy. Severe deficient 25(OH)D values in immigrant mothers and their newborns have been documented in Italy and in Mediterranean area [44,45].

Moreover, given that specific autoantibodies targeting β-cells appear in the early years of life, and afterwards a progressive loss of 70–80% β-cells by autoimmune destruction begins the clinical onset of T1DM, it can be assumed that the earlier ages of clinical onset represent more aggressive environmental pressures. In this regard, in two wide Italian series by the SIEDP (Italian Society of Pediatric Endocrinology and Diabetes), a mean earlier onset of T1DM was found in immigrant children born in Italy than in those born in their original countries (in 2004, born in Italy, mean age at onset, 4.0 years [IQR 2.2–6.9] vs. those born in original countries, 7.9 years [IQR 5.1–10.7, *p* < 0.001], and in 2014, born in Italy, 5.1 years [IQR 2.2–7.7] vs. born in their original countries, 7.8 years [IQR 5.3–10.3, *p* < 0.001]) [46,47]. The bimodal shape of the curves of ages at T1DM onset in immigrants in Italy, according to the site of birth, were interpreted as a possible expression of deeper vitamin D deficiency in children born in Italy, in early infancy or in pregnancy, than in those born in their original countries. As most of the immigrants came from North Africa, especially Morocco, it was speculated that there was an involvement of maternal darker skin pigmentation, poor dietary vitamin D intake, the reduced sunshine exposure for clothing, and an inadequate vitamin D supplementation in mothers of babies born in Italy. Was also found that in Moroccan T1DM children living in Italy there was a deeper vitamin D deficiency than in those living in the Sahel region of Morocco [48].

Emerging data promote the utility of vitamin D in co-supplementation with *n*-3 PUFAs since the clinical onset of T1DM [49]. We observed a child who, from the onset of disease, was supplemented with vitamin D (1000 IU/day) and *n*-3 PUFA (60 mg/Kg/day, EPA (Eicosapentaenoic Acid) and DHA (Docosahexaenoic Acid) ratio 2:1). After one year of follow-up, this child showed a partial remission with persistent fasting of C-peptide (marker of endogen secretion of insulin), and a low daily insulin need (<0.05 IU/Kg/day) with optimal metabolic control [50,51]. Based on this finding, a case-control study was performed in all new T1DM onsets in 2017 in our center, comparing onsets of the previous two years [52]. Both groups were equally supplemented with vitamin D (cholecalciferol 1000 IU/day) and were on a Mediterranean diet, for which they had received the same educational training, but cases received a further dietetic assistance to reduce foods rich in *n*-6 PUFAs. After one year of supplementation of *n*-3 PUFAs (50 mg/Kg/day, EPA: DHA, 2:1), a decreased insulin demand (*p* < 0.01) was found, particularly as pre-meal boluses (*p* < 0.01), and a decreased insulin demand adjusted for HbA1c (glycosylated hemoglobin, index of metabolic compensation, *p* < 0.05). (Figure 1).

Finally, Stene et al., 2003, in a large population-based, case-control study in Norway [53] showed that cod liver oil (a strong source of *n*-3 PUFAs) supplementation during the first year of life was associated with a lower risk of childhood-onset T1DM (OR: 0.74; 95% CI: 0.56, 0.99). Norris et al., 2007, in Colorado (US) [54] in DAISY (Diabetes Autoimmunity Study in the Young), an observational study in children at increased genetic risk for T1DM, found that a dietary intake of *n*-3 PUFAs was associated with a reduced risk of islet autoimmunity (OR, 0.45; 95%, CI, 0.21-0.96; *p* = 0.04), and more recently in 2021, Löfvenborg et al., in a large European study (EPIC-InterAct Study) [55], found an association between incident diabetes and a combination of GAD65 antibody (the most common autoantibody marker of autoimmune diabetes of adulthood) and fish intake or *n*-3 PUFAs plasma levels. In this study, individuals with high levels of GAD65 (≥ 167 IU/mL) and low *n*-3 PUFAs plasma levels had a higher risk of incident diabetes (OR 4.26, CI 2.70–6.72). This suggests that the intake of dietary fish or high *n*-3 PUFAs levels may prevent or delay diabetes onset in GAD65 antibody-positive individuals.

In conclusion, pending further knowledge, we speculate that some nutritional interventions could assign advantages over T1DM.

Specifically, vitamin D supplementation (i) in immigrant children from Africa, possibly since infancy and already during pregnancy in their mothers, (ii) in the siblings of those with T1DM if vitamin D deficiency is found, (iii) at onset of T1DM according to the above-mentioned study of Treiber et al. While a level of vitamin D of 25(OH)D, 30 ng/mL is generally considered adequate for the prevention of skeletal diseases and CVD, the threshold for other diseases, including T1DM, should be higher, in the range of 40 or 60 ng/mL [56]. Those levels are safe, 100 ng/mL being the upper limit, and can mostly be achieved through supplementation [57,58].Assuming *n*-3 PUFAs as supplementation, or fish intake, (i) in individuals with GAD65 autoantibodies, (ii) at the onset of T1DM, when there is a proved deficiency, in co-supplementation with vitamin D, for favoring a partial remission, probably counteracting the process underlying the selective autoimmune destruction of β-cells. The *n*-3 PUFAs supplementation should be coupled with diet, increasing *n*-3 and lowering *n*-6 PUFAs intakes. In planning a clinical intervention on the shortage of *n*-3 PUFAs, the *n*-6/*n*-3 ratio is crucial. Given that *n*-3 PUFAs, *n*-6 PUFAs, and saturated fats compete for the common metabolic pathways regulated by the same enzymes for the synthesis of eicosanoids with opposite functions, the ratio *n*-6 PUFAs: *n*-3 PUFAs is the best reference, and it should be optimal, ≤4:1. [59]. Note that the intake of fish or fatty fish in the diet, the concentration of *n*-3 PUFAs in plasma or in red blood cells, or simpler AA: EPA (Arachidonic Acid *n*-6: Eicosapentaenoic Acid *n*-3 PUFAs) ratio, are quite different and non-interchangeable parameters for investigations. The evaluation of PUFAs levels in red blood cell membrane phospholipids possibly represents the most precise approach [60].

One point that remains open is whether these nutritional interventions represent a delaying onset or mitigation of the underlying autoimmune process of T1DM, a question that should be assessed through specific RCTs.

## 4. The Path to Personalized Nutrition for Youth with Type 1 Diabetes

Recent research points to glucose continuous monitoring (CGM) for categorizing glycemic response to diet, which should be introduced to personalized nutrition non-diabetic people [61], and prospectively also to those with type 1 and type 2 diabetes. Given that CGM assists the diabetic patient regarding real-time guidance of the glycemic trend, the downloading of data during outpatient visits allows physicians to generate an ambulatory glucose profile (AGP), which is usually a guide for the therapeutic changes of individual patients. In particular, AGP reports generate metrics such as glucose management indicator (GMI), time in range (TIR), and glycemic variability (GV) as degrees of metabolic compensation [9,62]. Some of the metrics of those reports (GMI, TIR, GV), established over CGM of 14 days, may be more useful for the clinical management of T1DM than glycosylated hemoglobin (HbA1c) reflecting 3 months of glycemic control. Therefore, comparing the consecutive two week periods, before and after a nutritional change, CGM metrics allow both patients and physicians to understand outcomes by nutritional modification, if there are otherwise unchanged conditions of physical activities, lifestyle, and diet. Moreover, in clinical conditions of relative stability of the pre-meal glycemia through an adequate basal insulin, the glycemic elevation induced by the meal is controlled by the bolus of insulin and its timing of administration, and most contribute to metabolic T1DM compensation. The evaluation of glycemia 2 h after the meal and the area under the curve (AUC) described by the glycemic elevation act as a reference for the adequacy of the pre-meal insulin bolus. In practical diabetes management, the predictability of the elevation induced by the components of the meal remains a challenge for physicians and patients. The time to peak (TTP), defined as the time taken to reach the highest recorded glucose value within three hours of eating, can concur in a simpler way to evaluate the impact of nutrients on glycemic outcome. Reasonably, TTP has given a new input to the personalized nutrition in adulthood [63]. Unfortunately, few data have been found so far for children and adolescents regarding this approach to personalized nutrition. Furthermore, in a study on youth, 8–15 years old, with T1DM, a substantial variability of TTP between individuals and intra-individual was found [64]. This scenario is challenging for children and adolescent T1DM management, and meal-induced glycemic instability accounts for much of the frustrations of young patients and their parents.

Counteracting the post-prandial glycemic instability could be the research goal that personalized nutrition should aim for. Because the intake of micronutrients, specifically *n*-3 PUFAs alone or in combination with vitamin D, are not accurately assessed, at least in child and adolescent T1DM, individual glucose variability should be tested. AGP and TTP after consuming meals should be utilized as adequate tools, both under controlled and free-living conditions, as was carried out for macronutrients [63,64].

One more clinical case in our experience gave further confirmation. It involved a girl with T1DM with severe diabetic ketoacidosis at onset without any remission, and C-peptide levels below the limit of the test. Four years beyond the clinical onset of the disease, this patient showed a metabolic benefit during *n*-3 PUFA supplementation (50 mg/Kg/day, EPA:DHA, 2:1). The comparationrison of GCMs during follow-up, 18 months before and 18 months after *n*-3 PUFA supplementation, saw a significant reduction of mean glucose and GV, and a significant increase in TIR (70–180 mg/dL), without significant hypoglycemics events [65] (Figure 2). An amelioration of glycemic levels was evident during day and night in fasting state, but not within prandial time, likely because adequations of insulin boluses at meals were performed (Figure 3). This report highlights the possible impact of a nutrient on a specific T1DM person evaluated from CGM data.

This experience suggests a possible path to personalized nutrition for T1DM people, which is assessable for individual subjects.

An open point is understanding the mechanism of *n*-3 PUFAs. PUFAs are generally recognized to play a determining role in inflammation, but in this subject, far from the onset of disease and in the absence of residual endogen insulin secretion, and at normal youth weight, a metabolic process is perhaps involved. Further investigations should be carried out on the metabolic effects of *n*-3 PUFA in diabetes.

## 5. Nutrition in the Type 2 Diabetes of Youths and Its Prevention

T2DM in childhood, adolescence, and adulthood is closely tied to overweight and obesity, and to an underlying excess of food intake. Since 2006, meta-analysis and systematic reviews have shown that it is a preventable disease [66,67], even if several interventions of theoretical efficacy have found limited effectiveness. Based on evidence and adequate trials, in order to perform successful therapies, diversified approaches should probably be introduced according to different ethnicities, habits, and cultural backgrounds [68]. Therefore, herein we will focus on Italian cases. Given the emerging Globesity [69], T2DM prevention stands as one of the greatest commitments of pediatricians and children’s diabetologists, through encouragement to avoid overeating and to favor motor activation [70]. The most recommended suggestion is to promote regular exercise for one hour a day for children and adolescents’ at a medium to high intensity. Proper nutrition and lifestyle must also be established from the first years of life onwards. Furthermore, a family history of T2DM in parents and/or of gestational diabetes must also alert pediatricians to a possible adverse outcome in a normal-weight child [71].

Dietary indications for T2DM in adolescents largely overlap those of adults. The Medical Diabetes Association (AMD) and the Italian Diabetes Society (SID) have recently released guidelines for adults with T2DM [72]. Both SEARCH (SEARCH for Diabetes in Youth study) [73] and TODAY (Treatment Opinions for Type 2 Diabetes in Adolescents and Youth) [74] have provided a summarized therapy in published RCTs for children and adolescents with T2DM. The EarlyBird cohort study showed a reduced β-cells function since 5 years of age in children with normal fasting glucose who developed impaired fasting glycemia (IFG) during puberty [75]. Moreover, there was evidence that T2DM appears more aggressive in youth than in adults, despite similar characteristics of insulin resistance and impaired β-cells function [75]. These characteristics in youth vs. adults are relatively recent and need to be better understood. To date, the implementation of correct lifestyle and nutrition from early childhood onwards are milestones in preventing β-cell dysfunction developing into open disease.

In overt T2DM, nutritional therapy in adolescents overlaps that of adults, such as in macronutrient intake. According to AMD/SID guidelines, the Mediterranean diet has greater benefits in terms of HbA1c and weight reduction, rather than a low CHO consumption diet [72]. Of note, in Italy the intake of CHO is on average greater than in other countries, so a reduction of carbohydrates would seem more difficult to achieve in Italian people. Instead, a recent RCT states that dietary carbohydrate restriction performs a greater glycemic-lipidic control with weight loss in adulthood [76]. There are no corresponding RCTs for T2DM of pediatric age. In a recent study in T1DM diabetic children, improvements of HbA1c and TIR were found according to adherence to a Mediterranean diet [77]. ISPAD provided a Clinical Practice Consensus Guidelines Compendium in 2014, as a reference for the diet and lifestyle of T2DM in children and adolescents [78].

The understanding of specific gene–nutrient interactions is theoretically investigable with the contribution of T2DM-associated SNPs loci in GWAS. This procedure represents a first step in identifying measurable phenotypes as markers of this complex disease (e.g., fasting plasma glucose, fasting insulin, proinsulin levels, HOMA index). Quantitative trait locus analysis (QTL), identifying the gene-traits recurrently associated with a specific phenotype, allows one to define which genetic traits are most frequently linked to the disease. Through this method, more than 500 gene variants (SNPs) were found to be associated with T2DM (http://www.genome.gov/gwastudies, accessed on 28 September 2022), of which only one group has been linked with the known phenotypes of β-cells dysfunctions. Furthermore, one point of misunderstanding of a specific determinant is the interaction between various environmental factors, mostly the physical activity that limits or abolishes the biological effect of a nutrient on the genetic predisposition [79]. To date, genome-wide data explain <20% of the risk, in the context of genetic factors, which approximately account for 50% in determining T2DM. Despite the fact that the gene-related knowledge of determinants is rapidly progressing, a precise therapy focused on specific phenotype markers of disease and SNPs has not yet identified factors that consistently improve outcomes in T2DM prevention or prediabetes [79]. Finally, given the heterogeneity of T2DM, with different disease subtypes, as shown by recent research [80], interactions between environment and genome are expected according to the subtypes, and they likely also differ in pathophysiology, clinical expression, and outcomes [81]. This last aspect of the research would point to a precision therapy tailored to the patient.

In conclusion, the prevention of T2DM relies on countering excessive caloric intake and promoting physical activity from infancy, which are the hallmark for pediatricians and diabetologists. The effective strategy of implementing nutritional education in the primary school and promoting physical activity and sports in young people is far beyond the mere duty of physicians, and it would involve health and social educational strategies. Nevertheless, the planning of an effective policy for the prevention and treatment of T2DM in the young, supported by the caregivers of children and adolescents, is needed in our country. Further aspects in nutrition that should be overseen by pediatricians are the changes that have taken place in Italy in the last decades that are possibly reversible determinants of T2DM. Consequently, the identification of gene–nutrition relationships worsening or improving glycemic-lipidic metabolism could be a further step in T2DM prevention from youth onwards.

## 6. Insight into the Protective or Causative Roles of Specific Nutrients in Type 2 Diabetes

The review by Bernà et al. summarizes a thorough analysis of investigating genetic variations according to dietary patterns [27]. To pursue an accurate insight into the role of a specific nutrient in youth T2DM, the following should be investigated in RCTs vs. controls: selected youth populations of adolescents with a specific known genotype, stable overweight, and steady physical activities and nutritional intakes; the only variable should be the specific nutrient intended for study.

The complexity of the procedure could be overcome if common pathways underlying the fat accumulation driving T2DM should be found. Pathways verified in adults causing obesity, metabolic syndrome (Met-S), and T2DM could be identified previously in youth. Possible factors in this role are dyslipidemia, hepatic steatosis, and imbalance of *n*-6/*n*-3 PUFA levels ratio. As a marker of process, insulin resistance could then assist clinicians in the follow-up of patients.

As an example, a decrease in hepatic steatosis was documented in cases of obese teenagers carrying the *PNPLA3* rs738409 genotype at risk of obesity, Met-S, and T2DM, with a dietetic reduction of the *n*-6/*n*-3 PUFAs ratio [82]. In this study, even in controls not carrying this genetic variant, favorable outcomes were found through the reduction of this ratio. These data suggest that lowering the *n*-6/*n*-3 PUFA ratio in the diet ameliorates the non-alcoholic fatty liver disease (NAFLD) of obese adolescents, and a reduction in high *n*-6/*n*-3 PUFA level ratio to normal 4:1 should be pursued to counteract NAFLD in obese youth.

The recent literature suggests that the Western diet’s imbalance between high *n*-6 and low *n*-3 PUFAs intake contributes to obesity, but there are conflicting results of trials on the use of *n*-3 PUFA to reduce weight [83]. Given that hepatic fat accumulation has a pivotal role in insulin resistance [84,85], the central unanswered question is whether or not *n*-3 PUFA supplementation plays a role in this specific pathway. It is certain that *n*-3 PUFAs decrease hypertriglyceridemia and limit the synthesis of LDL-cholesterol, but in the complex dyslipidemia underlying obesity [86] and T2DM [87], it is not proven.

Furthermore, *n*-3 PUFA supplementation should be entered as personalized nutrition in a patient, coupled with the reduction of *n*-6 PUFA in the diet, if a high *n*-6/*n*-3 PUFA ratio was found, referring to 4:1 as the target [88]. Finally, to pursue a balanced *n*-6/*n*-3 PUFA levels, it is important to counteract inflammation of the obese state [89].

Practically, given the opposite effects of *n*-3 and *n*-6, their level ratios must be of reference for clinicians. The *n*-6 and *n*-3 PUFAs determinations in red blood cell membrane phospholipids represents the most precise way for investigations [60,83]. Finally, it is important to consider that the ratio *n*-6/*n*-3 PUFA is not the only expression of nutritional contributions, because genomic investigations found different haplotypes of fatty acid desaturases, which are key enzymes in the synthesis of long chain PUFAs [83]. For example, the haplotype D of fatty acid desaturase limits the synthesis of eicosanoids, involving phenotypes characterized by increased fat mass, obesity, dyslipidemia, diabetes, and CVD risk.

Given the wide span of different clinical conditions underlying or coexisting with overweight, it is probable that better results are gained with an articulated path of interventions. Therefore, in the assessment of personalized nutrition, it is important to consider in each patient the degree of obesity and the concomitance of associated problems (such as NAFLD, dyslipidemia) that might refer to specific guidelines [90], and a possibly precise therapy according to GWAS incoming knowledges [91]. Insulin resistance is an indicative parameter of the process and is easy to assess. It can be calculated by homeostatic model assessment (HOMA-IR Index, calculate [glycemia mg/dL × insulinemia µUI/mL] / 22.5, normal value 0.23–2.5) [92].

For example, we expect that an *n*-6/*n*-3 PUFA balanced diet would have the most success in conditions of “metabolically healthy obesity”, as shown by the research of Rondanelli et al. [93]. In those adult overweight/obese patients, after two-months of a balanced diet, improvements of *n*-6 and *n*-3 PUFAs and other metabolic indices, including insulin resistance, were found. Instead, in obese youths carrying the *PNPLA3* rs738409 genotype with NAFLD, the intention of the precise therapy is to increase the intake of *n*-3 PUFA and to decrease *n*-6 PUFA, glucose, sucrose, and fructose derivates from soft drinks, with physical activation and, according to N. Santoro, with “the dear and old diet, always a fundamental and irreplaceable NAFLD therapy” [94].

Vitamin D is another micronutrient of interest in T2DM, Met-S, and prediabetes because it can interfere both in β-cell insulin synthesis and in its release through a rise in intracellular calcium concentration [95]. Serum 25(OH)D concentration in T2DM plays a determinant role if it is in an optimal range between 40 and 60 ng/mL, to counteract the evolution of prediabetes to T2DM [96,97]. Prospective studies have shown that lower 25(OH)D status is a risk factor for Met-S [96] and T2DM [98]. Moreover, the preeminent role of vitamin D level in reducing the risk of T2DM is reported in a meta-analysis of 16 prospective studies [99]. However, there is not a definite conclusion on the protective role of vitamin D towards T2DM. The D2d Research Group, in a large RCT, found similar hazard ratios of T2DM in subjects, prediabetes and not, within 2.5 years of administration of 4000 IU/day of cholecalciferol or placebo [100]. Nevertheless, obesity represents a condition which itself lowers 25(OH)D levels [101], enhancing inflammation and insulin resistance [102], which is a mandatory decrease with an adequate intake of vitamin D.

In the complex interplay between nutrients and genes, the intestinal microbiome probably plays major roles in mediating the biological effects of nutrients. Given that micro-organisms present in the intestinal tract promote the breakdown and digestion of food-derived fragments by fermentation, the composition of the gut microbiota is expected to determine the intestinal derived products. Significant amounts of short-chain fatty acids (SCFAs) are generated from the breakdown and fermentation of complex carbohydrates, such as acetate, propionate, and butyrate, which contribute to glucose and lipid metabolism. Butyrate and acetate are lipogenic, becoming substrates for cholesterol and PUFA synthesis, and propionate prompts gluconeogenesis [103]. SCFAs, moreover, influence host hormonal regulation through incretins, which are peptide hormones secreted by enteroendocrine cells, such as glucagon-like peptide-1 (GLP-1), glucose-dependent insulinotropic peptide (GIP), and pancreatic peptide tyrosine (PYY). These entero-hormones slow gastric emptying, increase levels of satiety, promote feeding behavior and weight control, reduce insulin resistance, and improve glycemic control [104].

What is significant is that both the obese state and T2DM are characterized by lowering *Firmicutes-to-Bacteriodetes* ratios [105,106]. What is interesting is that metformin, a common drug lowering glycemia, modulates microbiota, thus improving the *Firmicutes-to-Bacteriodetes* ratio, suggesting that the metformin effect is mediated by gut microbiota [107].

A possible approach to a cure for obesity and diabetes through microbiota manipulation is still in its infancy. The possibility of diversifying the intestinal bacterial flora with pro- and pre-biotic nutrients, and *n*-3 PUFAs supplementation to repair the microbiota, would be suggestive. Moreover, among infants genetically susceptible to T1DM, it was found that a reduced diversity of microbiota and an expanded *Bacteroidetes* representation characterized those who developed the disease compared to those who did not [108]. Consistently, also in T1DM children, the risk factors for T1D include higher *Firmicutes* levels (OR 7.30; IC 2.26–23.54), whereas having a greater amount of *Bifidobacterium* in the gut (OR 0.13; IC 0.05–0.34) was a protective factor for T1D [109]. Moreover, microbiota regulate the integrity of the intestinal barrier, preserving tight junctions’ proteins, which are crucial for avoiding endotoxemia (toxic pathogen-derived metabolites released into the blood), which potentially induce inflammation and activate lipo-synthesis [110]. In animal models and in humans, the supplementation of *n*-3 PUFAs produced protection from endotoxemia caused by chemotherapy [111], antibiotics [112], and necrotizing enterocolitis of pre-term neonates [113], probably inducing favorable microbiota. Fecal microbiota transplant is nowadays a promising tool to manipulate the microbiota, with encouraging results in obese adults [114]. Therefore, future evolution might be a specific microbiota from a donor to a patient as therapy.

In conclusion, in T2DM therapy and in its prevention, particular attention should be paid to total caloric intake and to the distribution of macronutrients and micronutrients in accordance with the Mediterranean diet model. The intake of fiber, vitamin D, and *n*-3 PUFA must be coupled with a reduction of saturated fat and *n*-6 PUFA in the diet. The correction of vitamin D deficiency must be supported, at least in overweight children and adolescents, with supplementations to achieve an optimal 25(OH)D level in the range of 40–60 ng/mL, which is safe and inexpensive. The *n*-6/*n*-3 PUFA ratio of 4:1, must be pursued primarily with diet, limiting both saturated fats and food rich in *n*-6 PUFA, and if necessary, with supplementation.

## 7. Diabetes Precision Therapy through Nutrients: Perspectives and Suggestions

Precision medicine is defined as “an emerging approach for disease treatment and prevention that takes into account individual variability in genes, environment, and lifestyle for each person” (Medline Plus. What is precision medicine?). Personalized nutrition aims to transpose the same objective in relation to nutrients, not necessarily linked to gene–nutrient relationships, which instead, according to the recent literature, is better identified by the adjective “precise” [115]. Personalized nutrition considers blood biomarkers, physical activity, gut microbiome, and diets to treat or prevent a disease.

Consequently, in the topic of this lecture, several nutrients could be entered as personalized nutrition of type 1 and type 2 diabetes, and their prevention.

A personalization process according more to pathways than to genes might (i) include the most patients (obese, T1DM, T2DM, prediabetes, IGT, IFG), (ii) find markers that can allow monitoring of the course, and (iii) achieve measurable endpoints [114]. The roadmap should then diversify the intensity of interventions related to the degree of the disease, and possibly refer to specific guidelines for contextual pathologies (i.e., NALFD, hypertension, dyslipidemia, and so on), and moreover, given a probable emerging understanding of gene–nutrient links, allow moving the patient to a “precise therapy” when possible. Finally, the effectiveness of personalized nutrition in diabetes should be assessed in specific patients, comparing data prior to and after nutritional modifications.

## Figures and Tables

**Figure 1 nutrients-14-04887-f001:**
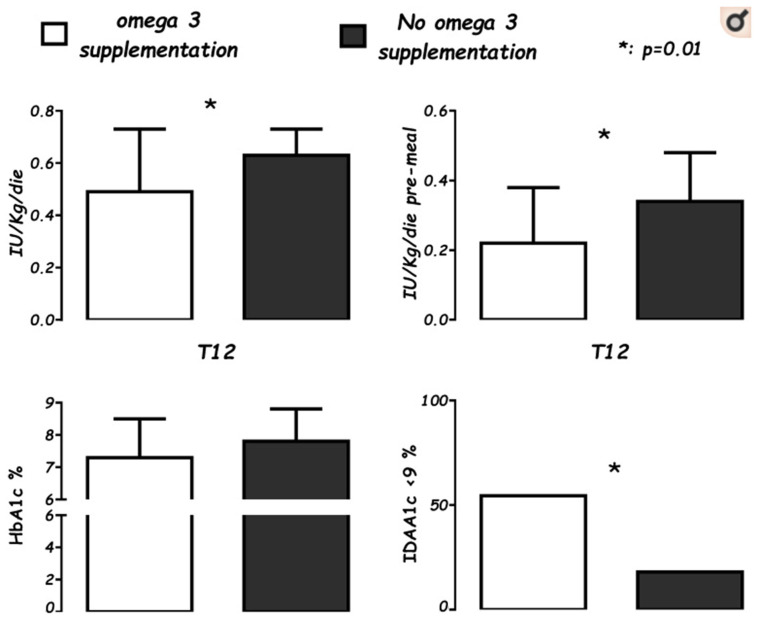
Levels of average daily insulin needs (IU/Kg/die), pre-meal boluses (IU/Kg/die), glycosylated hemoglobin percentage (HbA1c%), and the Insulin Dose Adjusted for glycosylated hemoglobin A1c percentage partial remission index (IDAA1c ≤ 9), in supplemented cases (white), for 22 patients after 12 months (T12) of *n*-3 supplementation vs. controls (black), a group of 37 not-supplemented patients at T12 from onset. (Cadario et al., Nutrients 2019; 11: 2158).

**Figure 2 nutrients-14-04887-f002:**
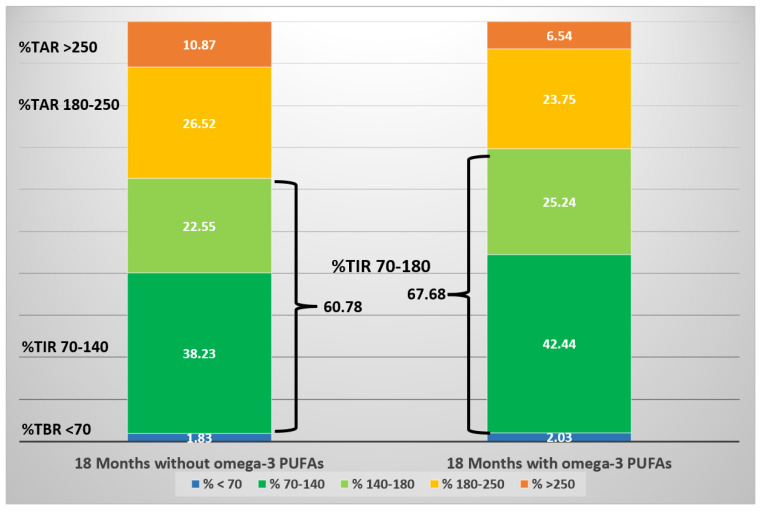
Schematic diagram of CGM metrics of an adolescent girl over three years, before and after the initiation of *n*-3 PUFA supplementation. The stacked bars represent the proportion of time (expressed as percentage) spent within a specific target glucose range. (Cadario et al., CellR4 2020; 8: e2879).

**Figure 3 nutrients-14-04887-f003:**
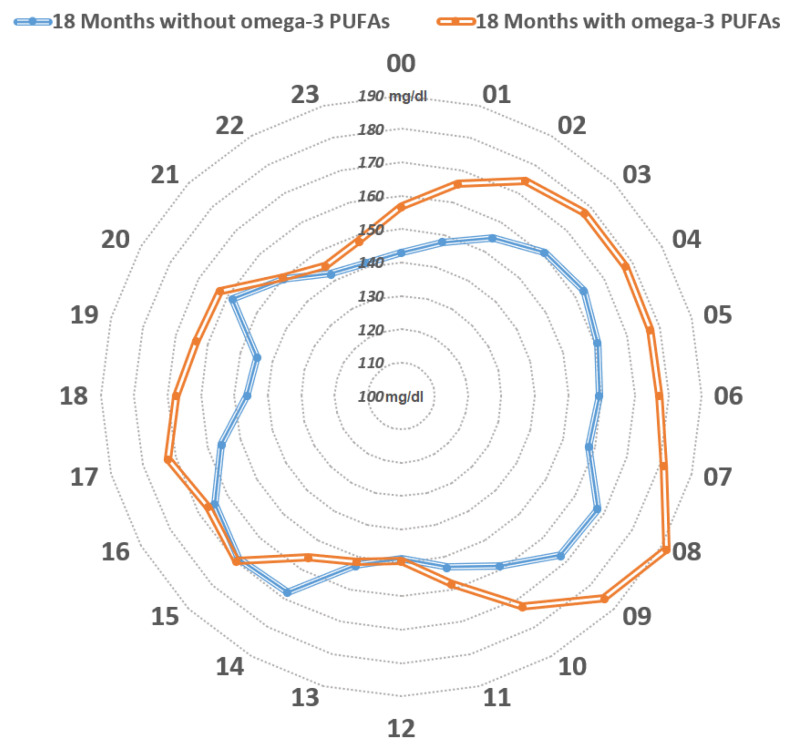
Distribution of the mean glucose values mg/dl over 24 h. The orange line represents mean glucose values within the 18-month-period prior to the initiation of *n*-3 PUFA supplementation, whereas the blue line represents mean glucose values during the 18-month-period following the initiation of *n*-3 PUFA supplementation. The blue line encircles a smaller area compared to the orange line. From midnight to midday, constant lower mean glucose values were observed upon *n*-3 PUFA supplementation, while glucose values appeared to be similar between mealtimes (lunch, afternoon snack, and dinner). (Cadario et al., CellR4 2020; 8: e2879).

**Table 1 nutrients-14-04887-t001:** From ISPAD Guidelines 2018 for children and adolescents [7], modified.

Carbohydrates must contribute 45% to 55% of daily energy intake
Moderate sucrose intake (up to 10% of total energy)
Fat—30% to 35% of total energy
<10% saturated fat + trans fatty acids
Proteins—15 to 20% of total energy

## Data Availability

Not applicable.

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
