# Peer review of "Vitamin D and ω-3 Polyunsaturated Fatty Acids towards a Personalized Nutrition of Youth Diabetes: A Narrative Lecture"

_nutrients, 2022, doi:10.3390/nu14224887_

Round 1

Reviewer 1 Report

The work of Cadario F.: “Vitamin D and omega-3 polyunsaturated fatty acids towards a personalized nutrition of young's diabetes: a narrative lecture” is an interesting overview of knowledge about T1DM and T2 DM in connection with nutritional habits of macronutrients and selected micronutrients as well as in connection with genetic malformations.

Dr. Cadario is certainly one of the most important pediatric diabetologists and his experiences are summarized in the presented work.

The work is divided into logical chapters concerning T1DM and T2DM in children and adolescents, but compares some markers with their values in adults.

I positively evaluate that after each chapter the author summarized the knowledge under "Conclsions" and also discussed open questions

The work was well read and will certainly be excellent literature not only for scientific teams, but also for practicing diabetologists.

However, there are minor inaccuracies in the work that I recommend to correct:

- mn-3 PUFA - correct probably as n-3 PUFA (line 67)

- the recommended amount of PUFAs 80 to 12 g also applies to vitamin D??? (lines 67-68)

- unite kg (lines 70, 71, etc) and Kg (for example line 192,242, in Figure 1) throughout the

   work

- explain Th17 and Treg (line 201) and what of them was reduced? It would be important for

    understanding, because of they have the opposite function with respect to the anti- and pro-

     inflammatory environment

- 60 mg/kg/day of n-3 PUFA - state the ratio of EPA and DHA (line 239, 333) - similarly

   Elsewhere, if it is given

- In the text to figure 1, insulin units are given as IU/Kg/day, but in the figure as IU/Kg/die –

   unify

- replace the word "cytokines" with "eicosanoids" (line 287)

- It is necessary to check the citation numbers in the text, as they do not correspond to the

   context, e.g. citation 60 (line 297), or 94 (line 487) is not the work of Rizzo, but of

    Matthews  et al, 1985. The work of Rizzo et al is in the references under number 60. And  

    some others !!!

- explain abbreviations, e.g. DKA (line 331), NAFLD (line 453)

Author Response

Thanks for your consideration, and more for your accurate revision. Please see the attachement.

  • mn-3 PUFA - correct probably as n-3 PUFA (line 67) corrected
  • the recommended amount of PUFAs 80 to 12 g also applies to vitamin D??? (lines 67-68)Thanks. I deleted "and vitamin D" to avoid confusion.
  • unite kg (lines 70, 71, etc) and Kg (for example line 192,242, in Figure 1) throughout the work done. 
  • explain Th17 and Treg (line 201) and what of them was reduced? It would be important for 

    understanding, because of they have the opposite function with respect to the anti- and pro-

     inflammatory environment

    I modified the text simplifying, and reporting only about Th17. The matter is really complex because the role of subsets Treg/ICOS+ habitually reduced in T1DM, and Treg ICOS-, were reduced after vitamin D supplementation for 6 months. In the context of reduced Th17, was supposed Treg/ ICOS+ aren't probably aging worsening the autoimmunity. Instead Th17 decrease after 6 months of supplementation

    - 60 mg/kg/day of n-3 PUFA - state the ratio of EPA and DHA (line 239, 333) - similarly

   Elsewhere, if it is given Done. thanks.

- In the text to figure 1, insulin units are given as IU/Kg/day, but in the figure as IU/Kg/die –

   unify Done

- replace the word "cytokines" with "eicosanoids" (line 287) Done

- It is necessary to check the citation numbers in the text, as they do not correspond to the

   context, e.g. citation 60 (line 297), or 94 (line 487) is not the work of Rizzo, but of

    Matthews  et al, 1985. The work of Rizzo et al is in the references under number 60. And  

    some others !!! Thanks. I ceeked all references and the citations in the text

- explain abbreviations, e.g. DKA (line 331), NAFLD (line 453). Done

Reviewer 2 Report

The ABSTRACT does not tell the reader what the paper is about. It is not even clear if this is a paper on T1D or T2D. Please rewrite it. 

Line 136 FF - tell us more about the Dutch Famine Study. You mention it without elaborating.

Line 198 - Th, Tregs - what are these? you need to explain.

Line 230 - from which countries were most of the immigrants - Africa? Middle East?

Line 271ff - sometimes you discuss PUFA, other times vitamin D.  Can you separate them more cleanly? In other words, write in studies examining the impact of nutrition on preventing/ameliorating T1D, two factors were examined: PUFA and vit D. The PUFA studies showed x and the vit D studies showed y

Line 294-325 - what is the point of all of this? It is not coming to a point or conclusion.

Line 315 - pike is peak.

Lines 330 - this case report seems to fit better in the prior section.

Lines 433-493 - can the information here be presented more succinctly? it is quite long and can be stated in fewer words.

Line 494 ff - a randomized prospective study did not find Vit D to prevent T2D. See NEJM Aug 8, 2019.

Author Response

Thanks for the accurate revision and for your suggestions. Please see the attachment

1)The ABSTRACT does not tell the reader what the paper is about. It is not even clear if this is a paper on T1D or T2D. Please rewrite it. 

Thanks for the remark. I submit now a revised Abstract, that may better resume the article

Line 136 FF - tell us more about the Dutch Famine Study. You mention it without elaborating. 

Done, see pages 3,4 in the text.

Line 198 - Th, Tregs - what are these? you need to explain.

I changed the text, according with the notice of the 1st Reviewer, simplifying and had added Th17 as lymphocyte subset, at page 5 

Line 230 - from which countries were most of the immigrants - Africa? Middle East?

I added in page 5, "mostly immigrants were from North Africa"

Line 271ff - sometimes you discuss PUFA, other times vitamin D.  Can you separate them more cleanly? In other words, write in studies examining the impact of nutrition on preventing/ameliorating T1D, two factors were examined: PUFA and vit D. The PUFA studies showed x and the vit D studies showed y

Excuse for the paragraph. I changed according to your suggestion.

Line 294-325 - what is the point of all of this? It is not coming to a point or conclusion.

Thanks for suggestion. I added a paragraph at page 8.

Line 315 - pike is peak.

Thanks, corrected.

Lines 330 - this case report seems to fit better in the prior section.

Thank you for your suggestion. However, I would prefer to insert this clinical case in the current context. This clinical case was the experience that made me aware of the potential of n-3 PUFA to reduce glycemic variability, and motivate myself to a personalized nutrition, as a scientifically evaluable path in the individual patient.

Lines 433-493 - can the information here be presented more succinctly? it is quite long and can be stated in fewer words.

Done. I hope fill you in accordance on the new version.

Line 494 ff - a randomized prospective study did not find Vit D to prevent T2D. See NEJM Aug 8, 2019.

Excuse for the important omission, I read the article, but then forgot to cite it. Thanks.
